Resource

# Cross-comparison of strains used for mitochondrial imaging in *Caenorhabditis elegans* during aging

Juri Kim* , Naibedya Dutta*, Matthew Vega, Andrew Bong, Maxim Averbukh, Rebecca Aviles Barahona, Athena Alcala, Jacob T Holmes, Gilberto Garcia, Ryo Higuchi-Sanabria

Measurements of mitochondrial morphology are a powerful proxy for assessing mitochondrial health, particularly during aging when organelle dynamics are disrupted. *Caenorhabditis elegans* provides an ideal system for in vivo mitochondrial imaging, but widely used high-copy transgenic strains can induce artifacts that confound interpretation because of their impact on cellular and organismal health and physiology. Here, we present and validate a suite of *C. elegans* strains expressing single-copy, matrix-localized GFP in the muscle, intestine, and hypodermis using the MosSCI technology. These strains enable robust, tissue-specific visualization of mitochondrial morphology without the caveats associated with multi-copy reporters. We benchmark their performance against existing models and demonstrate that our mitochondrial reporters are similarly capable of assessing age-associated mitochondrial morphology, while avoiding defects in cellular and physiological health associated with the multi-copy reporters. Furthermore, we assess how aging methods, bacterial diets, and inhibition of fusion and fission machinery impact mitochondrial morphology during aging. Our findings provide a standardized and physiologically relevant platform for studying mitochondrial dynamics during aging in *C. elegans*.

## Introduction

Mitochondrial fitness and function are critical for proper health and function of a cell because of their role in numerous cellular processes, including energy production, apoptotic and necrotic cell death regulation, calcium and amino acid storage, lipid oxidation, and heat production (Kamer & Mootha, 2015; Wang et al, 2023). Disrupting mitochondrial function can result in major consequences, including metabolic dysregulation, accumulation of toxic reactive oxygen species, and dysregulation of many cellular pathways (López-Otín et al, 2023; Zong et al, 2024). As such, mitochondrial dysfunction is one of 12 major biological hallmarks of aging, whereby mitochondrial dysfunction is observed during the natural aging process in most model organisms (López-Otín et al, 2023). Mitochondrial dysfunction can be defined by many measurable outcomes including loss of mitochondrial membrane potential, import, and respiratory capacity; accumulation of mitochondrial DNA mutations; loss of stoichiometry of multi-protein complexes in the mitochondria; and changes in mitochondrial morphology, mass, and volume (Amorim et al, 2022; Zong et al, 2024). Often, many of these features are correlated and occur simultaneously, which allows for the usage of one of these markers as a general readout for mitochondrial quality and function.

Mitochondrial morphology is a commonly used feature to indirectly determine mitochondrial function, as the quality control of mitochondria is regulated by mitochondrial dynamics, a tightly coordinated balance of continuous fusion and fission events that determine the shape, length, and number of mitochondria (Wang et al, 2023). The loss of balance between fusion and fission can have dramatic impacts on mitochondrial function, whereby excessive fission events or decline in fusion can lead to fragmentation of mitochondria, whereas excessive fusion or reduced fission can result in hyperfusion (de Boer et al, 2021). Generally, fragmentation and excessive fission of mitochondria are correlated with loss of membrane potential and loss of mitochondrial function (Zorova et al, 2018). This suggests that a shift toward fission is bad and that more fusion would be beneficial; however, excessive mitochondrial fusion can disrupt important quality control machinery, such as mitophagy that clears damaged mitochondrial components (Ashrafi & Schwarz, 2013). Thus, it is the collective balance of fusion and fission events that are important in maintaining cellular homeostasis, and abnormal mitochondrial dynamics can result in pathology of age-related diseases including cardiovascular diseases (Wu et al, 2019; Quiles & Gustafsson, 2022), cancer (Anderson et al, 2018), and lung disorders (Sharma et al, 2021). These studies highlight the importance of studying mitochondrial dynamics and visualizing their morphology during aging.

*Caenorhabditis elegans* serve as an excellent model system to study mitochondrial dynamics because of the low cost of their maintenance, established genome, and transparent body that allows for microscopic visualization of mitochondria in live animals.

---

Leonard Davis School of Gerontology, University of Southern California, Los Angeles, CA, USA

Correspondence: ryo.sanabria@usc.edu
*Juri Kim and Naibedya Dutta contributed equally to this work

**Life Science Alliance**

Importantly, their short lifespans allow for large-scale aging studies whereby mitochondrial imaging can be performed throughout the lifespan of the worm. Finally, there are robust genetic tools available for genetic modifications of *C. elegans* including CRISPR/Cas9 genome editing (Dickinson et al, 2015) and RNA interference (RNAi) (Bosher & Labouesse, 2000), which allow for identification of novel genetic mechanisms that impact mitochondrial dynamics and aging. Importantly, the regulation of mitochondrial dynamics is highly conserved in *C. elegans*, and the structure and function of mitochondria are highly similar to those of mammalian cells. Mitochondrial fusion in *C. elegans* is controlled by the conserved inner and outer membrane fusion proteins, EAT-3 (ortholog of Opa1) and FZO-1 (ortholog of Mfn1/2) (Rolland et al, 2009). Fission is controlled by the dynamin-related protein DRP-1 (ortholog of Drp1), which constricts the mitochondrion to separate mitochondrial membranes (Labrousse et al, 1999).

One of the most common methods to visualize mitochondria in *C. elegans* is to use genetically encoded mitochondrial-localized fluorophores because of the ease of genetic manipulation in this model. However, many of the currently existing methods involve transgenic animals with high-copy expression of fluorophores, including a mitochondrial matrix–localized green fluorescent protein (hereafter referred to as MLS::GFP) (Hoffmann et al, 2009) or overexpression of a red fluorescent protein (RFP)–tagged mitochondrial-localized protein, such as TOMM-20 (Wang et al, 2018). The benefit of these high-copy expression constructs is that because the fluorophores are expressed at very high levels and thus very bright, low-sensitivity cameras and weak excitatory light sources can be used to robustly visualize mitochondria. However, with the advent in technological advancement in microscopy and sensitivity of cameras in the past few decades, there is no longer a need for such high expression of fluorescent molecules for detection. Importantly, there are many potential caveats of high-copy expression, including a potential stress to the mitochondria to import so many proteins into the mitochondria (Begelman et al, 2022). In fact, while preparing this article, another group has independently identified that currently used methods suffer from several physiological caveats, including a significant reduction in lifespan (Valera-Alberni et al, 2024). In their article, the Mair laboratory illustrates the advantages and disadvantages of currently available tools to image mitochondria and offer a suite of single-copy mitochondrial membrane–localized fluorophores and endogenously tagged mitochondrial proteins as alternative strategies.

In our study, we offer another alternative strategy for mitochondrial imaging using a single-copy, matrix-localized fluorescent protein. Here, we used MosSCI transgenics for precise, stable, and single-copy expression of MLS::GFP in a known genetic locus. We compare and contrast our imaging strategies with the most commonly used strains in an attempt to standardize methods to image mitochondria in the muscle, intestine, and hypodermis in *C. elegans*. Importantly, our strains are complementary to the strains developed by the Mair laboratory and can be used to simultaneously visualize the outer membrane and mitochondrial matrix.

## Results

### Development of single-copy MLS::GFP strains using MosSCI in *C. elegans*

Mitochondrial morphology is often directly correlated with mitochondrial fitness and function and thus has gained popularity as the first line of study for understanding mitochondrial organization and quality under distinct circumstances. *C. elegans* serve as an exceptional model system for visualization of mitochondrial morphology, as the clear body allows for imaging of mitochondria in whole, live animals. However, current technologies for visualization of mitochondrial morphology have several distinct caveats: first, it used integration of a multi-copy MLS::GFP or TOMM-20::mRFP construct using a *myo-3* promoter for muscle-specific expression. These constructs are thus integrated into a random locus in the genome and have very high-copy expression of these fluorescent proteins, which could potentially impact mitochondrial quality and organismal physiology. Indeed, a recent study showed that these strains had measurable changes in longevity, reproduction, animal size, and generation time (Valera-Alberni et al, 2024). Moreover, these animals exhibit highly variable expression in fluorescence across tissues even within the same animal, making comprehensive studies and quantitative imaging very challenging.

Here, we sought to make more simplified versions of these strains by expressing MLS::GFP using the MosSCI system (Frøkjær-Jensen et al, 2012) to eliminate several caveats of previously used methods. These animals have MLS::GFP integrated into a known locus, which simplifies genetic crosses and allows for controlled, equal expression of the fluorophore across the entire animal. By fusing the MLS of ATP-1 to GFP and using cell type–specific promoters, we created robust methods to visualize mitochondrial morphology specifically in the muscle (*myo-3p*), intestine (*vha-6p*), and the hypodermis (*col-19p*) (Fig 1A).

To test the dynamic range of these reporters for visualizing mitochondrial morphology, we exposed animals to RNAi knockdown of genes encoding the fusion and fission machinery, *fzo-1* and *drp-1*, respectively. As expected, knockdown of fusion resulted in significant fragmentation of mitochondria (Figs 1B and 2). However, RNAi knockdown of *drp-1* resulted in aggregation of mitochondria, which is consistent with previous findings (Labrousse et al, 1999) that argue that hyperfusion of mitochondria results in formation of aggregated mitochondria that resemble spheres, which is also reflected quantitatively as a decrease of mitochondrial length (Figs 1B and 2). To better measure the dynamic range of fragmented versus fused mitochondria, we performed titration of *fzo-1* and *drp-1* knockdown and found that dilution of the *drp-1* RNAi with an empty vector (EV) RNAi to 20% (i.e., a 1:4 ratio of *drp-1: EV*) was optimal to block mitochondrial fusion and create a more tubular and interconnected structure in the muscle and intestine, rather than forming hyperfused spheres (Figs 1C and 2). However, in the hypodermis, 20% *drp-1* still resulted in mitochondrial spheres. A similar dilution of *fzo-1* RNAi to 20% still effectively fragmented the mitochondria in all tissues, but to a lesser extent than undiluted RNAi (Fig 1C).

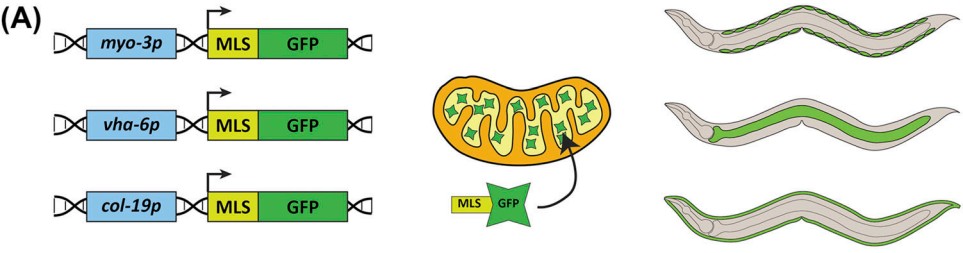

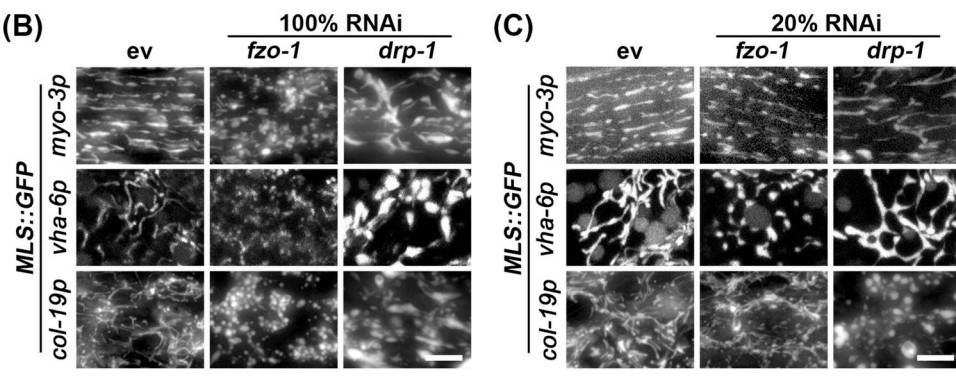

**Figure 1. Validation of strains by altering mitochondrial morphology using *fzo-1* or *drp-1* RNAi treatments.**
**(A)** Schematic of MLS::GFP, which we express using cell type–specific promoters: *myo-3p* for the muscle, *vha-6p* for the intestine, and *col-19p* for the hypodermis. MLS::GFP is imported into the matrix of the mitochondria and can be robustly visualized in live animals using fluorescent microscopy. **(B)** Animals expressing cell type–specific MLS::GFP were grown on control empty vector (ev), *fzo-1*, or *drp-1* RNAi from the L1 stage and imaged on day 5 of adulthood. **(C)** *fzo-1* and *drp-1* RNAis were diluted to 20% with ev (i.e., 1:4 ratio of RNAi:ev). Animals were grown on the indicated RNAis from the L1 stage and imaged at day 5 of adulthood. Scale bar is 10 μm.

To quantify mitochondrial morphology, we used mitoMAPR, which quantifies numerous metrics of mitochondria, including mitochondrial length, junction points (Zhang et al, 2019). Although mitoMAPR is able to robustly quantify the reduction in mitochondrial object length with age, there are some limitations when quantifying mitochondrial morphology in fusion and fission mutants. Previous literature has shown that skeletonization of images, which is required for quantification via mitoMAPR (Zhang et al, 2019), results in artifacts when quantifying fusion and fission mutants (Viana et al, 2015; Lewis & Marshall, 2023), which we also experienced and could not achieve robustly reproducible quantification of fusion/fission mutants when using mitoMAPR (Fig 2). However, some conditions did still show significance when using mitoMAPR. For example, knockdown of *drp-1* in the hypodermis displays a significant increase in mitochondrial length both at day 1 and at 9 (Fig 2C). However, *fzo-1* RNAi did not show a significant increase in object length in any tissue. However, when using the measurement of object number/mitochondrial footprint, which refers to the total number of mitochondrial objects detected and the total area occupied by mitochondria within a defined region of interest, this measurement showed a significant increase in the intestine and hypodermis of *fzo-1* knockdown. Fusion mutants should have more fragmented mitochondria, which should result in more distinct objects being recognized, as well as having these objects fill up more space in the cell. Therefore, object number could potentially be a good option to use for quantification of fusion mutants. Overall, care must be taken when using skeletonization methods for quantification of mitochondrial morphology.

In this study, we mounted worms directly onto HistoBond adhesion slides using M9 buffer, as previously described (Kim et al, 2025), differing from other methods that involve placing worms on agar pads. Animals placed directly onto slides have lower mobility compared to agar pads and reduces the need for use of chemicals that paralyze the worm. By comparing mitochondrial morphology after placing the worms directly onto slides with placing the worms on agar pads, we confirm that mounting worms directly onto the slides for a short time does not interfere with mitochondrial morphology (Fig S1). These data provide direct evidence that our MosSCI strains allow for robust visualization of mitochondrial morphology and behave as expected when mitochondrial dynamics are altered. Moreover, our data provide relative RNAi concentrations of *fzo-1* and *drp-1* that allow for alterations of mitochondrial morphology.

## Mitochondria exhibit fragmentation during aging

Next, we performed mitochondrial imaging during aging. Consistent with previous reports (Sharma et al, 2019), we see that animals display an increase in mitochondrial fragmentation during the aging process in all tissues (Fig 3A). Comparison of our MosSCI muscle-specific MLS::GFP strain with the most commonly used multi-copy integrated muscle mitochondrial strains using MLS::GFP and TOMM-20::mRFP showed that our strains exhibit a delayed fragmentation of mitochondria during aging (Fig 3B). This is likely due to the potential detrimental effects of multi-copy strains having to import a large quantity of mitochondrial-localized proteins (Begelman et al, 2022). Here, mitoMAPR showed a significant reduction in object length in the muscle from day 1 to 9, although at day 13 or in the multi-copy integrated MLS::GFP strain where mitochondria are severely fragmented, it failed to accurately quantify mitochondrial morphology, similar to fusion mutants. Similar to muscle mitochondrial imaging, high-copy expression of MLS::GFP in the intestine also resulted in premature mitochondrial fragmentation during aging compared with our MosSCI intestine-specific

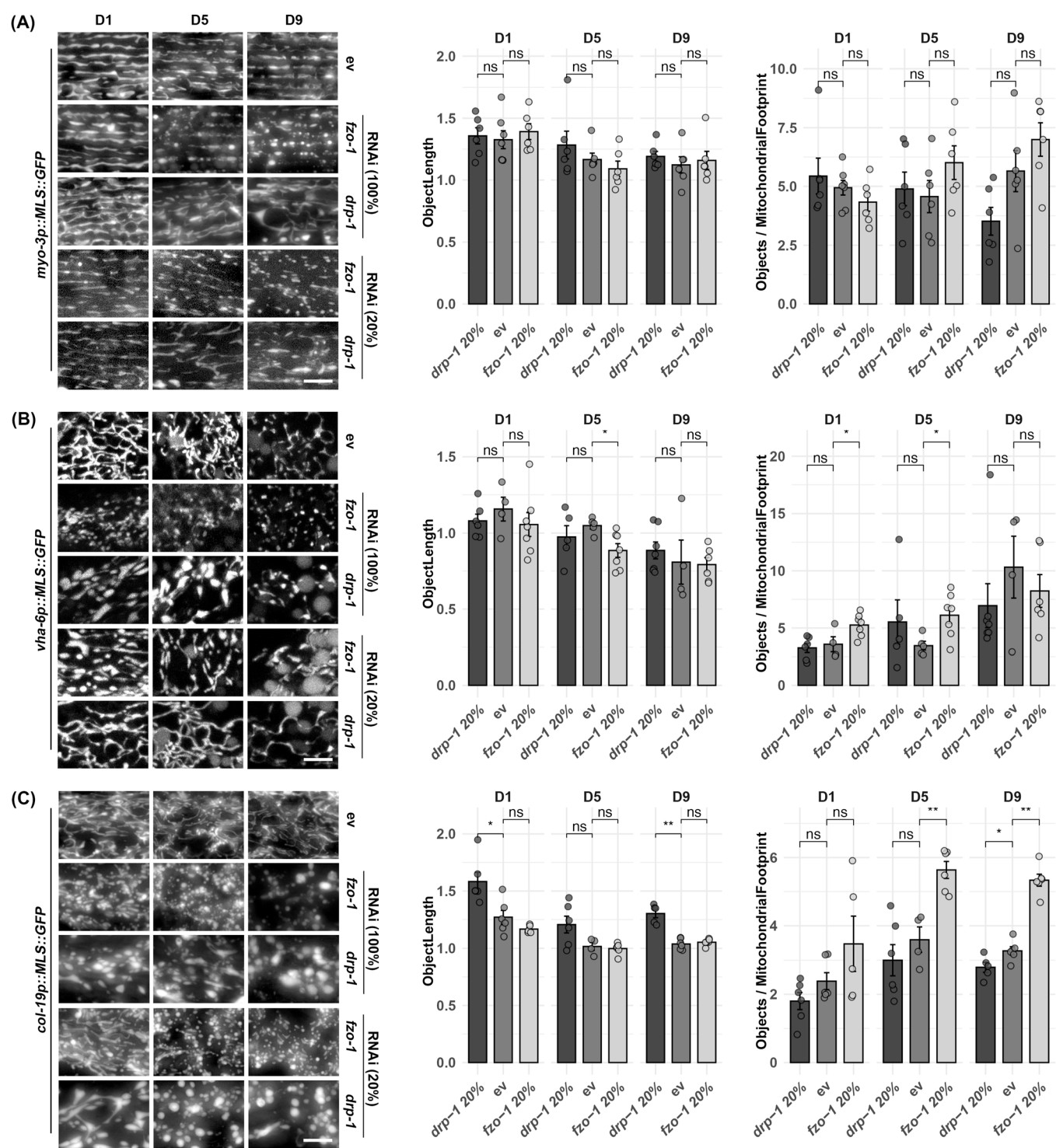

**Figure 2. Imaging of mitochondrial morphology during aging upon *fzo-1* and *drp-1* RNAi.**
**(A, B, C)** Animals expressing *myo-3p::MLS::GFP* (A), *vha-6p::MLS::GFP* (B), and *col-19p::MLS::GFP* (C) were grown on full concentration RNAi of control (ev), *fzo-1*, or *drp-1* from the L1 stage and imaged during days 1, 5, and 9 of adulthood. *fzo-1* and *drp-1* RNAis were treated 100% or diluted to 20% with ev (i.e., 1:4 ratio of RNAi:ev). Animals were grown on the indicated RNAis from the L1 stage and imaged at days 1, 5, and 9 of adulthood. All quantification was performed using mitoMAPR. Object length and objects/ mitochondrial footprint are shown here as example measurements, and all mitochondrial measurements measured by mitoMAPR are available in Table S1. In mitoMAPR-based quantification, the "objects/mitochondrial footprint" refers to the total number of objects detected as mitochondria and the area occupied by all the mitochondria within a defined region of interest. ns = not significant, $P > 0.05$, *$P < 0.05$, **$P < 0.01$ using a Wilcoxon signed-rank test. The scale bar is 10 $\mu$m.

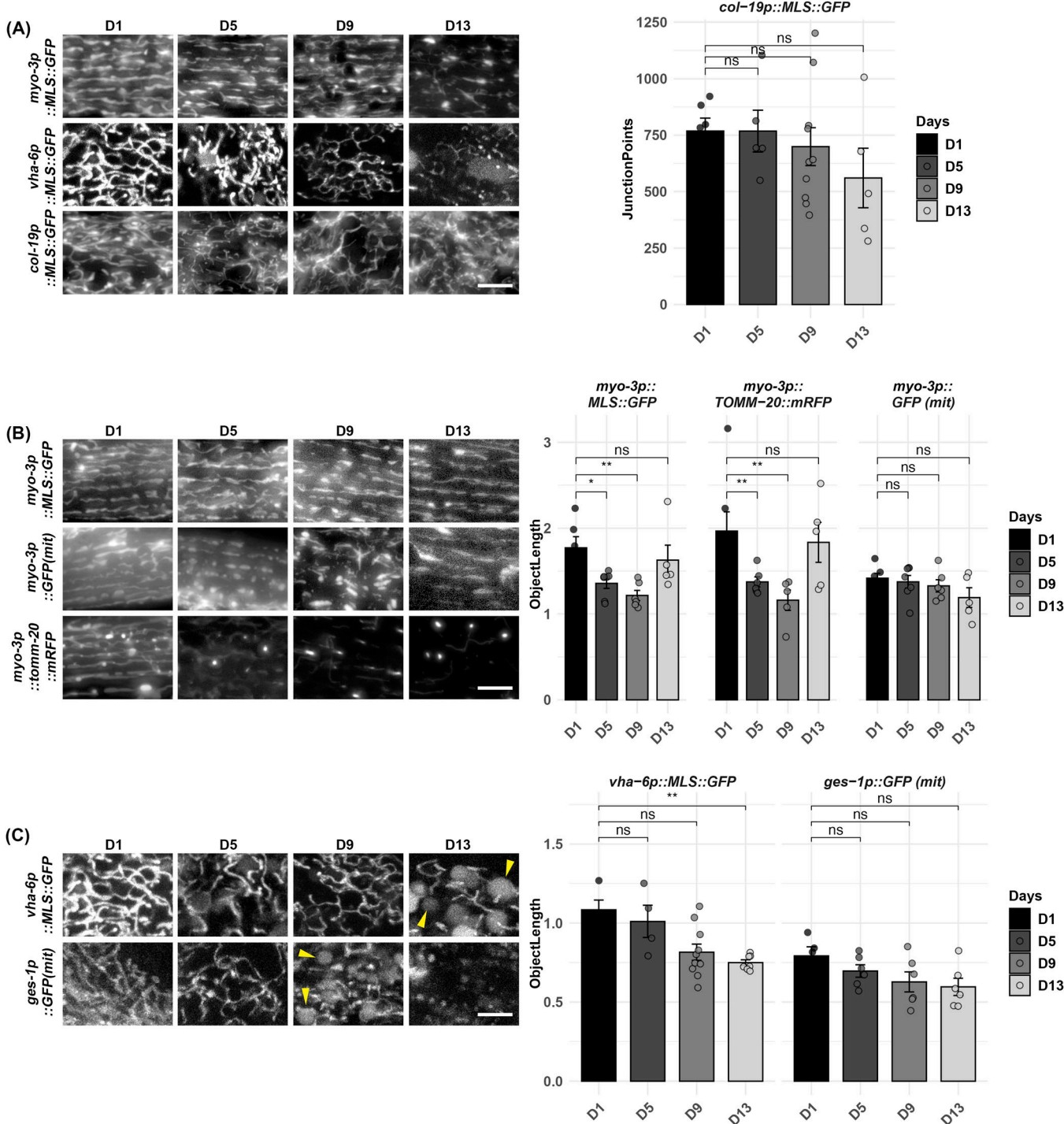

**Figure 3. Comparison of strains for cell type–specific imaging of mitochondria during aging.**
**(A)** Imaging of cell type–specific mitochondria in the muscle (*myo-3p*), intestine (*vha-6p*), and hypodermis (*col-19p*) using a single-copy integration of MLS::GFP using MosSCI during aging. **(B)** Comparison of muscle MosSCI MLS::GFP (RHS191) strain with multi-copy *myo-3::GFP(mit)* (SJ4103) and multi-copy *myo-3p::TOMM-20::mRFP* (PS6192) during aging. **(C)** Comparison of intestine MosSCI MLS::GFP (RHS193) with multi-copy *ges-1p::GFP(mit)* (SJ4143) during aging. All animals were grown on ev from the L1 stage and imaged at days 1, 5, 9, and 13 of adulthood. Arrowheads indicate examples of spherical autofluorescent structures in the intestine. All quantification was performed using mitoMAPR. Object length is shown here as an example measurement, and all mitochondrial measurements measured by mitoMAPR are available in Table S1. ns = not significant, $P > 0.05$, *$P < 0.05$, **$P < 0.01$ using a Wilcoxon signed-rank test. The scale bar is 10 μm.

MLS::GFP strain (Fig 3C). It is important to note that with intestinal MLS::GFP lines, we observed round fluorescent structure that increased in an age-dependent manner (Fig 3C, arrows). To further validate this is not a mitochondrial-related structure or artifact of our strain, we imaged animals expressing MLS::GFP in the intestine with DAPI excitation (365-nm laser), which should not excite the GFP fluorophore. We found that these round structures were visible with both DAPI and GFP excitation lasers, confirming this is not related to mitochondria (Fig S2), and likely an age-associated increase in autofluorescence as previously described (Komura et al, 2021).

During the preparation of this article, another group independently identified similar caveats of these multi-copy strains used to visualize mitochondrial morphology (Valera-Alberni et al, 2024). Importantly, we confirmed the Mair laboratory's findings that previously used strains exhibit significant variability in expression across cells, whereas our MosSCI strains showed consistent expression across cells (Fig S3A). In their study, the Mair laboratory also created novel single-copy strains that either use a minimal MLS from the TOMM-20 protein fused to a fluorescent molecule or directly integrate a fluorescent tag at the endogenous gene locus of TOMM-50 or TIMM-70. Similar to the strains presented here, their study eliminated many of the caveats from previously used strains, and are complementary to our imaging strategies. Because our fluorescent probes are localized to the mitochondrial matrix, it allows for imaging of the mitochondrial lumen, which can be directly paired with the outer membrane marker from the Mair laboratory to visualize multiple subcompartments of the mitochondria simultaneously (Fig S3B). This is an important consideration for those interested in dynamics of inner and outer mitochondrial membrane fusion and fission, which does not always occur simultaneously (Malka et al, 2005). An interesting phenomenon observed by the Mair laboratory was that when TOMM-20 or TIMM-50 were fused to RFPs, these proteins aggregated at old age, which did not occur when GFP was used. Similar to their findings, we also observe aggregation of the matrix-localized RFP, MLS::mRuby at old age, which we did not observe in any of our matrix-localized GFP strains (Fig S3C). This suggests that these RFP-related artifacts are not restricted to membrane proteins and also affect matrix-targeted proteins.

Although C. elegans offer a simple and easy way to study mitochondrial morphology during aging because of their short lifespan and ease of growth, one challenge is that they exist as hermaphrodites with the ability to self-fertilize. Therefore, for aging studies, progeny must be eliminated to prevent contamination of the aging cohort with their offspring. One common method of eliminating progeny is to chemically sterilize animals using exposure to 5-fluoro-2'-deoxyuridine (FUDR), which causes developmental arrest in progeny by preventing DNA replication (Bell & Wolff, 1964; Rooney et al, 2014). However, FUDR may have unwanted effects on aging and some studies have shown that exposure to FUDR can potentially impact the aging process (Mitchell et al, 1979; Rooney et al, 2014). Therefore, we compared exposure to FUDR with the standard method of daily manual picking of adults away from their progeny. Although we find that exposure to FUDR and manual picking of adults away from their progeny both show similar age-induced fragmentation of mitochondria, FUDR-exposed animals exhibited a slight delay in mitochondrial fragmentation in all cell types, although these differences were not statistically significant when quantified using mitoMAPR (Fig 4A–C). Importantly, this difference was not entirely due to the manual manipulation of worms. Animals exposed to FUDR, but manually moved daily, still exhibit a delay in mitochondrial fragmentation in muscle, but not in the intestine and hypodermis (Fig S4A–C). These data show that although exposure to FUDR may cause a minor delay in mitochondrial fragmentation with age in some tissue, there are no major artifacts induced by this aging method overall, and thus, FUDR exposure is likely a feasible approach to aging out animals for studies of mitochondrial morphology.

As an alternative approach to aging animals for those who wish to avoid FUDR exposure, another commonly used aging method includes using temperature-sensitive mutants including the germline mutant glp-4(bn2) (Beanan & Strome, 1992; Castro Torres et al, 2022) or the sperm-deficient mutant fer-1 (Ward & Miwa, 1978) and CF512 strain (Garigan et al, 2002). Here, the germline mutant glp-4(bn2) previously validated to not impact aging (TeKippe & Aballay, 2010) was crossed into our MosSCI MLS::GFP animals. Animals were grown at 22°C for the duration of their lifespan, as this elevated temperature was sufficient to sterilize animals in our hands. We found that glp-4(bn2) displayed more pronounced fragmentation in all tissues at all timepoints even from early ages, though it was not reflected in the quantification in some tissues when mitochondrial fragmentation was severe, similar to fusion mutants described above (Fig 4). Thus, this must be taken into consideration when using glp-4(bn2) animals for mitochondrial imaging studies.

Finally, we tested the impact of the bacterial food source on mitochondrial morphology during aging. Escherichia coli B strain OP50 and K strain HT115 are the most common food sources for C. elegans with OP50 being the most common food choice for standard maintenance and HT115 used for RNAi experiments (Reinke et al, 2010; Revtovich et al, 2019). However, previous work has shown that mitochondrial health is improved in worms fed an HT115 diet, likely because of increased availability of vitamin B12 (Revtovich et al, 2019). This is an important consideration, because mitochondrial morphology can exhibit significant differences based on the bacterial diet (Revtovich et al, 2019; Neve et al, 2020). Consistent with previous reports, we found that animals grown on OP50 exhibit more fragmented mitochondrial morphology compared with animals grown on HT115 in the muscle, intestine, and hypodermis (Fig 5A–C). Interestingly, many of these differences are most prominently observed at day 1 of adulthood, but differences were also noticeable during mid-age and old age. To determine whether these differences were due to differences in vitamin B12 as previously described, we performed mitochondrial imaging in animals grown on OP50 diets supplemented with a vitamin B12 analog (adoCbl; adenosylcobalamin, final concentration of 12.8 nM). We found that supplementation of adoCbl rescued the mitochondrial fragmentation in OP50 on day 1 of adulthood compared with the HT115 food source in all tissues. Moreover, the supplementation was even shown to delay the age-associated mitochondrial fragmentation of old worms (day 9 of adulthood) both on HT115 and on OP50 (Fig 5). Interestingly, adoCbl supplementation showed more pronounced interconnection of the mitochondrial network because there are increased junction points in all tissues after adoCbl supplementation throughout all the ages (Table S1). However, it is important to note that these quantified changes did not reach statistical significance. Altogether, our data presented

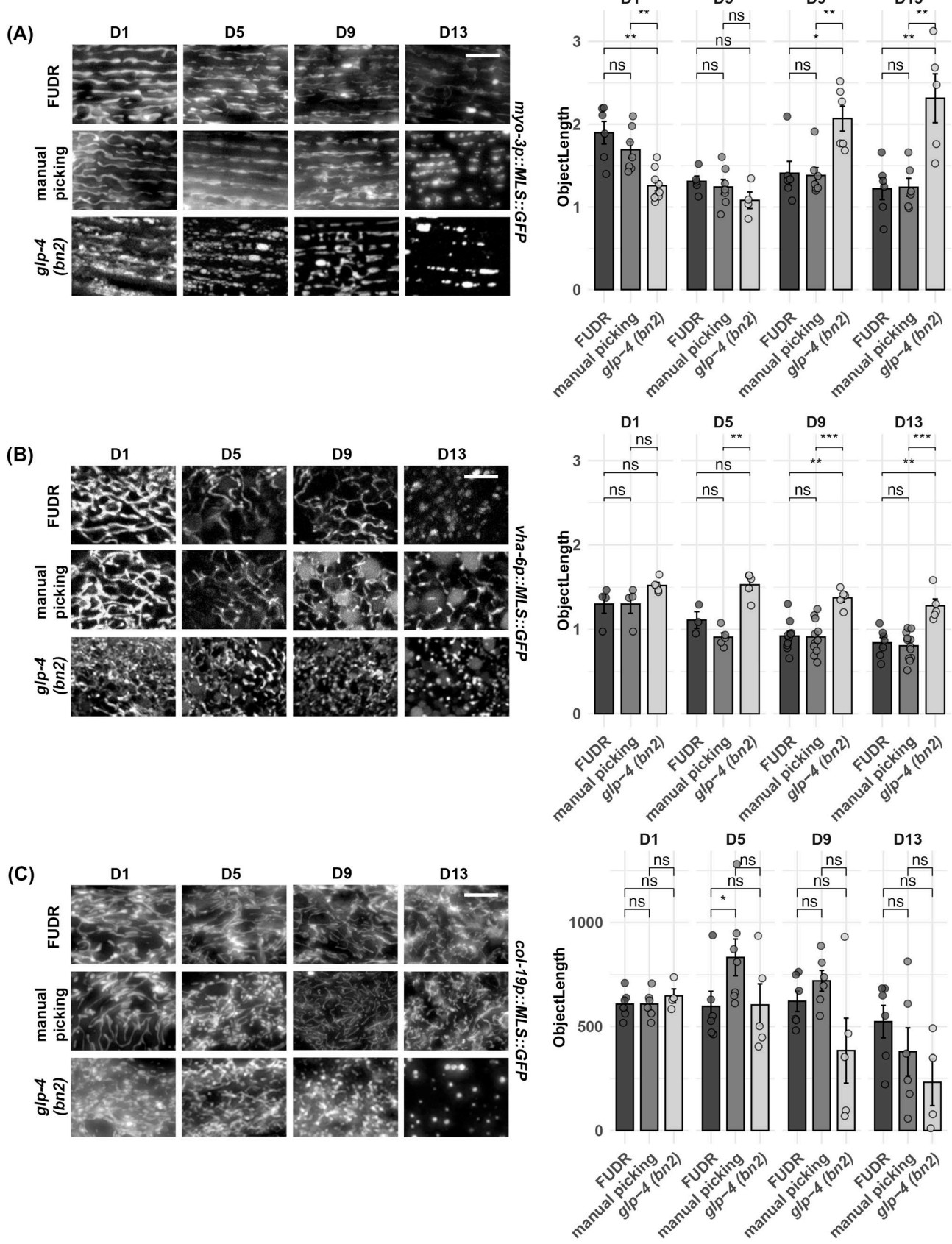

here provide evidence that our MosSCI-generated strains are robust and reliable reporters for mitochondrial morphology during aging. More importantly, although some differences exist in terms of methodology for animal growth or aging, there are no dramatic differences between strategies and consistency in using a single method—or testing multiple methods—are both viable options for aging experiments.

### MosSCI-generated single-copy MLS::GFP strains exhibit mild physiological changes

Because the most commonly used multi-copy MLS::GFP and TOMM-20::mRFP strains exhibited changes to several physiological measurements including longevity (Valera-Alberni et al, 2024), we next sought to characterize our MosSCI MLS::GFP strains for potential changes in organismal health and longevity. The multi-copy MLS::GFP strains showed minimal changes to lifespan, although one biological replicate out of four total replicates showed a mild decrease in lifespan (Figs 6A and S5A–D and Table S2). To further assess animal health, we measured locomotor behavior and saw no change in thrashing rates throughout aging in any MLS::GFP strains (Fig 6B). Finally, to more carefully evaluate mitochondrial function, we measured oxygen consumption rate (OCR) using a Seahorse assay. Interestingly, we found that all MLS::GFP strains exhibited a lower basal OCR compared with WT animals (Fig 6C). To ensure that this was due to a reduction in mitochondrial respiration, we measured OCR after treatment with sodium azide, which completely shuts down mitochondrial respiration and saw no difference between the MLS::GFP strains and a WT control, suggesting that the decrease in OCR is indeed due to a decline in mitochondrial respiration (Fig 6D). Overall, our data show that our MLS::GFP strains are not completely benign and may have a mild impact on mitochondrial respiration, but do not dramatically impact longevity or healthspan unlike the previously developed multi-copy strains (Valera-Alberni et al, 2024).

## Discussion

Imaging of mitochondrial morphology is a robust and simple method to get a general idea of mitochondrial quality, as changes to morphology are often correlated with changes to numerous functional measurements of mitochondria (Sharma et al, 2019; Chen et al, 2023). *C. elegans* serve as a robust model to perform mitochondrial imaging during aging, as its short lifespan and small, clear body allow for imaging of mitochondrial morphology throughout the entire lifespan of the worm in adult animals. However, there are many different methods to image mitochondrial morphology in the worm, each with its distinct advantages and disadvantages. Conventional mitochondrial dyes like MitoTracker and TMRE are great options because they do not require strain

construction, but suffer from variability in the amount of staining across cells, tissues, and individual animals, especially in the *C. elegans* model where the thick cuticle prevents entry of many dyes (Presley et al, 2003; Wang et al, 2016; Valera-Alberni et al, 2024). To circumvent this issue, researchers can deliver these dyes through their bacterial food source, but this method will not allow for robust or equal staining across all tissues (Ravi et al, 2021; Valera-Alberni et al, 2024). These caveats combined with the ease of genetic manipulation in the worm make genetically encoded fluorescent protein–based imaging strategies the most commonly used tools. However, even among fluorescent protein–based imaging, there are many different strains, each with individual advantages and disadvantages.

Here, we present a single-copy, matrix-localized GFP strain using a GFP bound to the MLS of ATP-1. Importantly, these transgenes were introduced using the MosSCI technology into a known genetic locus, thus preventing unwanted off-target effects of irradiation-based integration methods that integrate into an unknown locus and may interfere with the expression of important genes (Thellmann et al, 2003; Frøkjær-Jensen et al, 2008). In addition, we show that our low-copy constructs have limited effects on mitochondrial function and organismal health with only minor effects on OCR, unlike the high-copy expression strains that have numerous effects on mitochondrial function and have been shown independently by another lab to significantly affect whole organism physiology (Valera-Alberni et al, 2024). The MLS::GFP strains are complementary to the membrane-targeted fluorophores recently published by the Mair laboratory while this article was in preparation. The Mair laboratory used CRISPR/Cas9 technology to either endogenously tag mitochondrial membrane–localized proteins or express fluorophores with a minimal MLS of outer membrane proteins, which has similar benefits to our strains of single-copy expression in known genetic loci. Each of these strains also presents their own unique advantages; the Mair laboratory strains allow for visualization of mitochondria across the entire animal as it is ubiquitously expressed, whereas our strains allow for focusing on a single tissue as we used cell type–specific promoters. The Mair laboratory constructs allow for visualization of membranes, which have much higher resolving capacity to look at mitochondrial substructures, whereas matrix-localized fluorophores allow for accumulation of fluorescence in one area and thus are often brighter. Each strain can also be used for FRAP experiments where our strains will allow for measurements of mitochondrial matrix continuity, whereas the Mair laboratory strains are optimal for measuring membrane fluidity and dynamics. Finally, we also show that these strains can be used together to simultaneously visualize the matrix and outer membrane.

Even among the strains presented in both studies, there are considerations to be made. The Mair laboratory found that red fluorophores—regardless of their identity (i.e., mCherry, mScarlet, mRFP)—showed aggregation in the mitochondria at old age. We

**Figure 4. Comparison of aging methods for imaging of mitochondrial morphology.**
Animals were aged using the following methods: (1) adults manually picked away from progeny, (2) chemical sterilization with FUDR where 100 µl of 10 mg/ml FUDR was dropped onto the food source, and (3) temperature-sensitive *glp-4(bn2)* grown at the 22°C restrictive temperature. Animals were grown on ev from the L1 stage and imaged during days 1, 5, 9, and 13 of adulthood. **(A, B, C)** Imaging was performed for MLS::GFP expressed in the (A) muscle, (B) intestine, and (C) hypodermis. All quantification was performed using mitoMAPR. Object length is shown here as an example measurement, and all mitochondrial measurements measured by mitoMAPR are available in Table S1. ns = not significant, $P > 0.05$, *$P < 0.05$, **$P < 0.01$, ***$P < 0.001$ using a Wilcoxon signed-rank test. The scale bar is 10 µm.

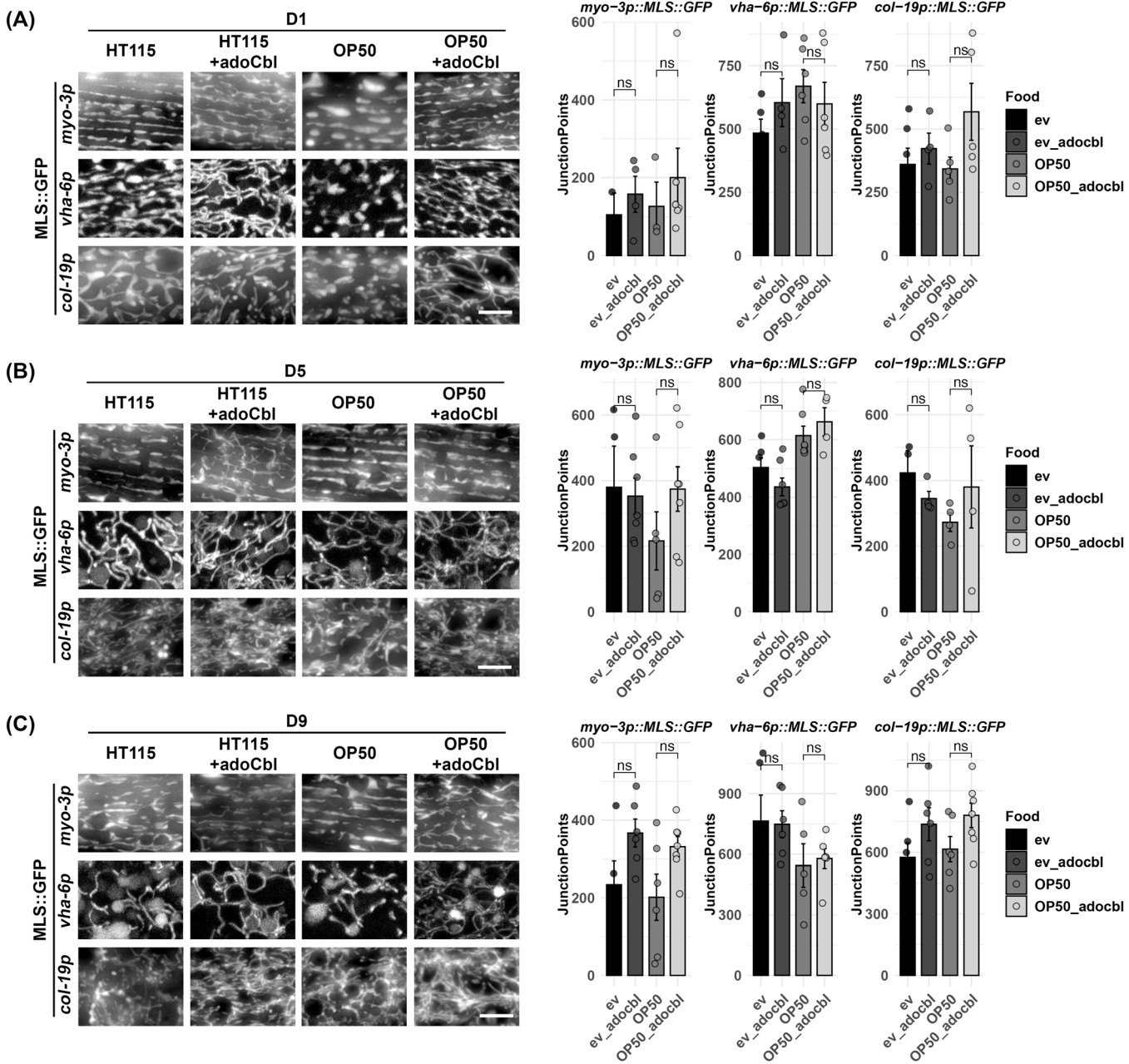

**Figure 5. Comparison of mitochondrial morphology on different diets.**
Animals were grown on an HT115 or OP50 diet with or without the vitamin B12 analog adoCbl. Animals were grown on ev from the L1 stage and imaged during days 1, 5, 9, and 13 of adulthood. **(A, B, C)** Imaging was performed for MLS::GFP expressed in the (A) muscle, (B) intestine, and (C) hypodermis. All quantification was performed using mitoMAPR. Junction points are used here as an example measurement, and all mitochondrial measurements measured by mitoMAPR are available in Table S1. ns = not significant using a Wilcoxon signed-rank test.

confirmed these findings using our matrix-localized mRuby constructs, adding evidence that this aggregation is common across multiple red fluorophores and that it is not limited to just mitochondrial membranes. This is an important consideration as aggregation of proteins both inside and at the outer membrane of mitochondria can result in induction of mitochondrial stress (Berendzen et al, 2016). Thus, some researchers may choose to avoid red fluorophores when regulation of mitochondrial protein

homeostasis is the primary area of study. However, red fluorophores are not the only ones that display caveats. Green and blue fluorescent proteins can cause issues because of autofluorescence in these channels in the *C. elegans* intestine (Komura et al, 2021). These autofluorescent structures appear as spherical structures that can obscure mitochondria, and thus, care must be taken to ensure that structures visualized using green or blue fluorophores in the intestine are truly mitochondrial structures. This is

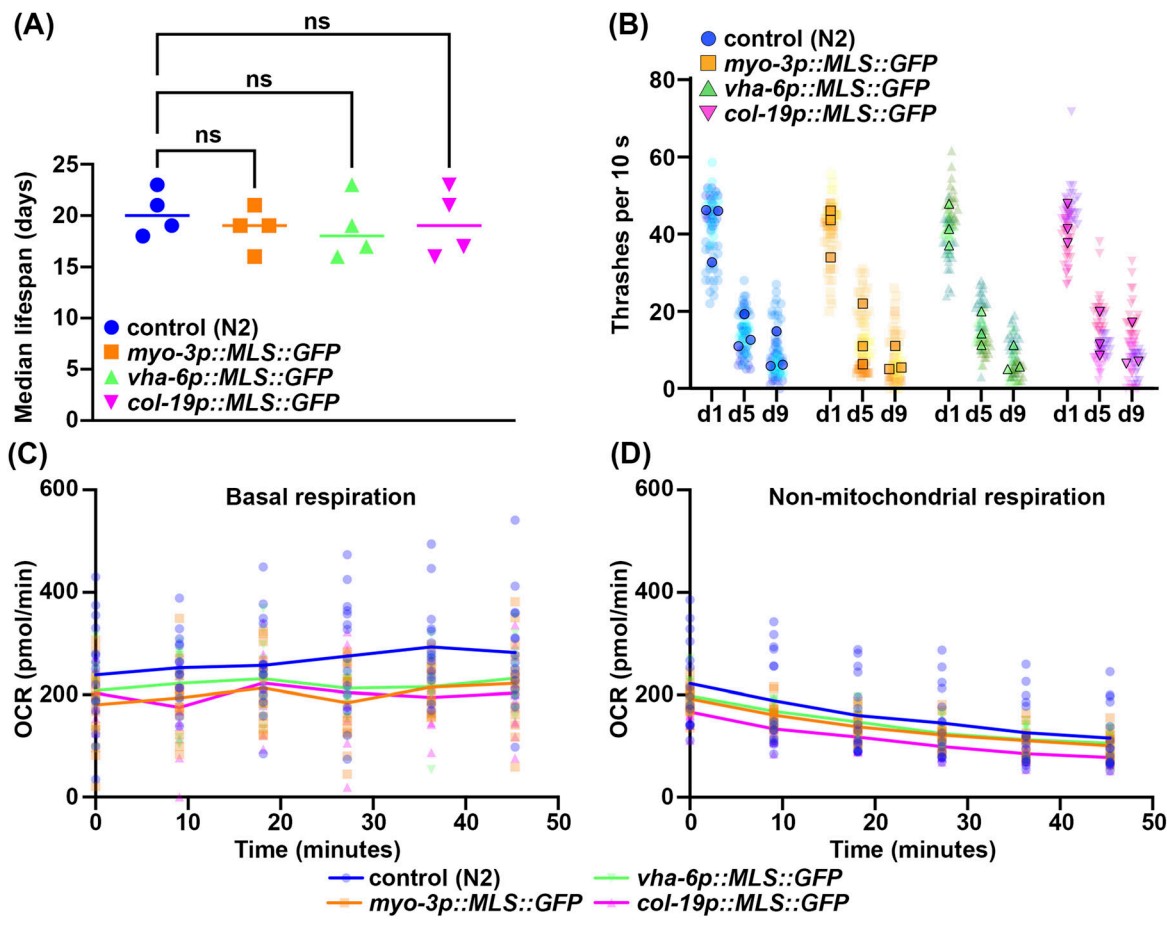

**Figure 6. Expression of MLS::GFP has very mild impacts on animal physiology.**
**(A)** Median lifespans of N2, *myo-3p::MLS::GFP, vha-6::MLS::GFP, col-19p::MLS::GFP*. Each dot represents the median lifespan of a replicate with worms of n > 90. (N = 4)
**(B)** Thrashing measurements were performed on N2, *myo-3p::MLS::GFP, vha-6::MLS::GFP, col-19p::MLS::GFP* grown on ev from the L1 stage and scored on days 1, 5, and 9 of adulthood. n = 20 for each strain per replicate. N = 3. All three replicates are superplotted, and the mean values of each replicate are indicated with outlined symbols.
**(C)** Basal respiration of n > 50 worms measured at day 1 of adulthood across six timepoints in 9-min interval in M9 solution. N = 3. **(D)** Nonmitochondrial respiration of n > 50 worms measured at six timepoints after 50 mM sodium azide treatment in 9-min interval. N = 3. Statistics by one-way ANOVA, using GraphPad Prism 10.0. ns = not significant, *P < 0.03, **P < 0.002, ***P < 0.0002, ****P < 0.0001.

even more important when using automated quantification software like mitoMAPR. This software often cannot distinguish mitochondria from autofluorescence and thus may present issues with quantification. Automated software like mitoMAPR that skeletonizes mitochondria into lines to perform quantification also has some caveats when quantifying fusion and fission mutants (Viana et al, 2015; Lewis & Marshall, 2023). Similarly, we found that mitoMAPR could not always accurately quantify mitochondrial length of severely fragmented mitochondria.

In addition to red and green fluorophores, blue fluorescent proteins have commonly been shown to have problems for mitochondrial imaging and perturb cellular health (Higuchi-Sanabria et al, 2016), potentially because of high levels of ROS production (Alvarez et al, 2010). Although some laboratories have prioritized identification of nontoxic blue fluorophores (Mohapatra et al, 2019), they have yet to be tested in the *C. elegans* system.

Although we focused primarily on strain choice in this article, there are also many additional important considerations for imaging of mitochondrial morphology. First, bacterial food choice is

critical as previous reports have shown that mitochondrial morphology may be different when animals are grown on the two most standard bacterial food choices, OP50 and HT115, because of deficiencies in vitamin B12 (Revtovich et al, 2019). The K strain *E. coli* HT115 is most commonly used for RNAi as the two largest RNAi libraries—the Vidal and Ahringer libraries—were constructed in this RNAi-competent strain (Ahringer, 2006). Subsequently, modifications were made to the B strain *E. coli* OP50 to make them RNAi-competent by deleting the RNAIII RNase and genomically introducing an IPTG-inducible T7 RNA polymerase (Neve et al, 2020). However, no thorough RNAi library exists yet in this construct and utility of this strain requires cloning each individual gene of interest into an RNAi vector and transforming it into this modified OP50 strain. Thus, for large-scale RNAi studies, the usage of HT115 is still unavoidable. However, our studies show that during aging, there are no major differences between the OP50 and HT115 diet in terms of mitochondrial morphology; simply, the OP50 diet displays slightly more fragmentation of mitochondria, but the trends for age-associated fragmentation are robustly apparent in either

diet. Moreover, as previously shown, mitochondrial morphology can be matched between OP50 and HT115 diets, solely by the addition of vitamin B12 in the OP50 diet, and we have found this to be true across the lifespan of the animal. Thus, unless major metabolic pathways are being tested where either excess vitamin B12 or other still uncharacterized differences between OP50 and HT115 diets may cause unmanageable confounding variables, we believe that standard mitochondrial imaging during aging is not extremely sensitive to differences in these standard diet choices. However, care should be taken by each researcher to confirm this for each of their experimental conditions.

Beyond metabolic differences between OP50 and HT115, researchers should also consider the usage of antibiotics. For HT115 bacteria, growth on tetracycline is often used to select for the correct bacteria as the RNase III allele (*rnc:14::ΔTn10*) in HT115 bacteria confers tetracycline resistance (Neve et al, 2020). Moreover, the pL4440 vector often used as the expression vector for dsRNA for RNAi experiments carries an ampicillin resistance gene, which researchers often select for using either ampicillin or the more shelf-stable carbenicillin. However, previous research has shown that exposure to specific antibiotics—including tetracycline (Chatzispyrou et al, 2015) and ampicillin (Khan et al, 2014)—can impact mitochondrial function. Although these studies may argue that usage of antibiotics should be avoided, this can be challenging in some cases as RNAi requires selection of plasmid-carrying bacteria and some common laboratory contaminants have been shown to impact many criteria of organismal health and longevity (Stuhr & Curran, 2023). Importantly, our study has already highlighted how OP50 and HT115 conditions are not dramatically different, especially when vitamin B12 differences are corrected for. Because our HT115 growth conditions include exposure to both carbenicillin and tetracycline while OP50 growth conditions do not, we can likely extrapolate these data to suggest that there are no major concerns for using antibiotics for the specific mitochondrial imaging conditions used in this study.

To further add to complications, other technical components can impact mitochondrial imaging. For example, a standardized method needs to be used to synchronize and age *C. elegans* populations. The most common method of synchronization is to perform an egg prep by bleaching of animals using a sodium hypochlorite solution; however, this applies some stress to the animals, which can impact metabolism (Verdú et al, 2022). As an alternative, egg-laying methods (Castro Torres et al, 2022) or manually picking animals at definable stages such as visibility of the L4 "crescent"-shared pre-vulva can be used, but these assays are generally manually intensive and would be challenging for large-scale assays. As an alternative, a commercially available *C. elegans* synchronizer can be used to harvest large volumes of L1 animals from a mixed population (Rasmussen & Reiner, 2021), but this may be cost-prohibitive for some laboratories. To compound this issue, once a synchronized population is established, one must also identify a method of choice for aging out cohorts of animals. Here, we provide three methods to age out animals: first, animals can be manually manipulated away from progeny. Although this is the most "natural" method to age out animals and does not require any interventions, this is also the most manually intensive and is more challenging for large-scale experiments. We also tested both chemical and genetic sterilization

techniques and found that they do not dramatically change mitochondrial morphology with age, although the timing may shift slightly. This shift in timing can be due to technical aspects, such as lack of manual manipulation of worms when using FUDR, which can reduce physical stress on the animals and delay mitochondrial fragmentation. Or in the case of using a temperature-sensitive mutant, the elevated temperatures may serve as a stress on the worm that can accelerate mitochondrial aging.

Overall, there are many considerations to be made when performing mitochondrial imaging in a laboratory, particularly during the aging process. Therefore, care must be taken to standardize methods for mitochondrial imaging in each laboratory, or proper controls must be performed using multiple methods to confirm that phenotypic findings are not artifacts of methods. Finally, although mitochondrial morphology can be used as an indirect measurement of mitochondrial function because morphology often correlates with mitochondrial function, there are many exceptions to this correlation (Osellame et al, 2012). Therefore, to perform a comprehensive analysis of mitochondrial health and function, additional measurements need to be made, including measurements of mitochondrial membrane potential (Chen, 1988; Sakamuru et al, 2016), ATP synthesis capacity (Zong et al, 2024), calcium levels (Pivovarova & Andrews, 2010), respiratory capacity (Zong et al, 2024), and mitochondrial DNA sequence and content (Castellani et al, 2020). However, as these assays can be technically challenging, imaging of mitochondrial morphology can be used as a first step in determining whether any experimental conditions affect general mitochondrial biology.

## Materials and Methods

### *C. elegans* strains and maintenance

All strains used in this study are derived from the N2 WT animal from the Caenorhabditis Genetics Center (CGC) and are listed in Table 1. Animals are maintained at 15°C on OP50 *E. coli* B strain bacteria on standard NGM (Nematode Growth Medium, 1 mM $CaCl_2$, 5 μg/ml cholesterol, 25 mM $KPO_4$, 1 mM $MgSO_4$, 2% agar wt/vol, 0.25% Bacto Peptone wt/vol, 51.3 mM NaCl) plates. Animals are maintained by either chunking a small patch of worms or manually picking a small population of young (before L4) animals onto a freshly seeded plate. Animals are only kept for ~25–30 generations in this way before thawing a new batch to avoid genetic drift.

For all experimental purposes, animals are age-matched using a standard bleaching protocol as previously described (Bar-Ziv et al, 2020). Briefly, animals are collected using M9 solution (22 mM $KH_2PO_4$ monobasic, 42.3 mM $Na_2HPO_4$, 85.6 mM NaCl, 1 mM $MgSO_4$) and bleached using a 1.8% sodium hypochlorite and 0.375 M KOH solution. After bleaching animals, eggs are washed 3–4x with M9 solution with repeated centrifugation at 1,100*g* and aspiration of solution. Intact eggs were floated in M9 solution in a rotator overnight at 20°C to obtain tighter synchronization at the L1 stage. Synchronized L1 animals were subsequently plated on RNAi plates (1 mM $CaCl_2$, 5 μg/ml cholesterol, 25 mM $KPO_4$, 1 mM $MgSO_4$, 2% agar wt/vol, 0.25% Bacto Peptone wt/vol, 51.3 mM NaCl, 1 μM IPTG, and 100 μg/ml carbenicillin,

**Table 1. Strains used in this study.**

| *C. elegans*: Bristol (N2) strain as WT | CGC | N2 |
|---|---|---|
| *C. elegans*: RHS19: *glp-4(bn2)* | CGC | SS104 backcrossed 6x |
| *C. elegans*: RHS191: *uthSi17[myo-3p::MLS::GFP::unc-54 3'UTR::cb-unc-119(+)] I; unc-119(ed3) III* | Moehle et al (2021) | AGD1664 backcrossed 4x |
| *C. elegans*: RHS192: *uthSi83[col-19p::MLS::GFP::unc-54 3'UTR::cb-unc-119(+)] I; unc-119(ed3) III* | Moehle et al (2021) | AGD2837 backcrossed 4x |
| *C. elegans*: RHS193: *uthSi80[vha-6p::MLS::GFP::unc-54 3'UTR::cb-unc-119(+)] I; unc-119(ed3) III* | Moehle et al (2021) | AGD2805 backcrossed 4x |
| *C. elegans*: RHS213: *zcIs[ges-1p::GFP(mit)]* | CGC | SJ4313 backcrossed 3x |
| *C. elegans*: RHS218: *uthSi17[myo-3p::MLS::GFP::unc-54 3'UTR::cb-unc-119(+)] I; unc-119(ed3) III; glp-4(bn2)* | This study | |
| *C. elegans*: RHS243: *uthSi83[col-19p::MLS::GFP::unc-54 3'UTR::cb-unc-119(+)] I; unc-119(ed3) III; glp-4(bn2)* | This study | |
| *C. elegans*: RHS217: *uthSi80[vha-6p::MLS::GFP::unc-54 3'UTR::cb-unc-119(+)] I; unc-119(ed3) III; glp-4(bn2)* | This study | |
| *C. elegans*: RHS218: *glp-4(bn2) I; uthSi17[myo-3p::MLS::GFP(65C)::unc-54 3'UTR, cb-unc-119(+)] I;* | This study | |
| *C. elegans*: RHS180: *wbmIs98[eft-3p::tomm-20(aa1-49)::mCherry::unc-54 3'UTR]; wbmIs65[eft-3p::3XFLAG::dpy-10 crRNA::unc-54 3'UTR]; uthSi17[myo-3p::MLS::GFP(65C)::unc-54 3'UTR, cb-unc-119(+)] I;* | This study | |
| *C. elegans*: PS6192: *syIs243[myo-3p::TOMM-20::mRFP + unc-119(+) + pBS Sk+]* | CGC | |
| *C. elegans*: SJ4103: *zcIs14[myo-3p::GFP(mit)]* | CGC | |
| *C. elegans*: SJ4143: *zcIs17[ges-1p::GFP(mit)]* | CGC | |
| *C. elegans*: AGD2319: *unc-119(ed3) III; uthSi62[vha-6p::MLS::mRuby::unc-54 3'UTR, cb-unc-119(+)] IV;* | Tharp et al (2021) | |
| *C. elegans*: AGD2883: *unc-119(ed3) III; uthSi90[myo-3p::MLS::mRuby::unc-54 3'UTR cb-unc-119(+)] IV;* | This study | |
| *C. elegans*: RHS243: *uthSi83[col-19p::MLS::GFP(65C)::unc-54 3'UTR, cb-unc-119(+)] I; glp-4(bn2) I;* | This study | |

with HT115 *E. coli* K strain containing pL4440 vector control or pL4440 with RNAi of interest) unless otherwise noted. All aging experiments were performed on plates supplemented with 100 $\mu$l of 10 mg/ml FUDR spotted directly on the bacterial lawn unless otherwise noted.

For growth of *glp-4(bn2)* animals, we grow animals at 22°C from the L1 stage. Although previous reports have shown that *glp-4(bn2)* animals are sterile at 25°C (Beanan & Strome, 1992), after backcrossing animals 6x to our N2 animals we found that our *glp-4(bn2)* animals were fully sterile at 22°C. Therefore, we opted to grow animals at 22°C to reduce caveats of potential induction of stress at 25°C (Gouvéa et al, 2015).

For vitamin B12 supplementation assays, NGM plates were supplemented with 12.8 nM of adenosylcobalamin, vitamin B12 analog. Adenosylcobalamin was added to the media post-autoclaving. Animals were grown on adenosylcobalamin-containing plates from the L1 stage throughout their lifespan.

### Making *C. elegans* transgenic strains

Transgenic *C. elegans* strains were generated using the Mos1-mediated Single-Copy Insertion (MosSCI) technique, following the detailed protocol described by Garcia et al (2022). Specifically, the transgenic strains used in this study were created by injecting MosSCI-specific strains with a plasmid cocktail. This cocktail included a plasmid vector with a transgene for Mos1 transposes (pCFJ601), a transgenic construct designed for the tissue-specific expression of MLS::GFP or MLS::mRuby within a MosSCI-compatible vector, and multiple tissue-specific co-injection fluorescent markers. Details of the injection cocktail and MosSCI-specific strains are provided in Tables 2 and 3, respectively.

In addition, the transgenic MLS::GFP *C. elegans* strains were crossed with *glp-4(bn2)* mutant animals. The *glp-4(bn2)* mutation in the resulting strains was confirmed through sequencing. Single-worm lysis was performed by placing single worms into dH$_2$0, proteinase K, and PCR buffer (we used Q5 PCR buffer) at a 42:3:5 ratio. The worm mixtures were then heated to 60°C for 1 h and 98°C for 20 min. 1 $\mu$l of this lysate was used as a template DNA for a standard PCR using the forward primer tgacataccattgaggcttgag and the reverse primer gtaaattgaccttggttgaggc. Standard Sanger sequencing was performed at Genewiz using the forward primer.

**Table 2.  MosSCI injection cocktail.**

| Plasmid | Transgene | Function | Expression tissue | Working concentration |
|---|---|---|---|---|
| pCFJ601 | eft-3p::Mos1 transposase | Transposase | Ubiquitous | 50 ng/µl |
| pGH8 | rab-3p::mCherry::unc-54 UTR | Co-injection marker | Pan-neuronal | 10 ng/µl |
| pCFJ90 | myo-2p::mCherry::unc- 54 UTR | Co-injection marker | Pharynx | 2.5 ng/µl |
| pCFJ104 | myo-3p::mCherry::unc- 54 UTR | Co-injection marker | Body wall muscle | 5 ng/µl |
| pCFJ35X | construct, Cb-unc-119(+) | MosSCI construct | Construct-dependent | 25 ng/µl |

**Table 3.  MosSCI strains used in this study.**

| Strain name | Genotype | Promoter | MosSCI vector | MLS gene | MosSCI strain | Integration site |
|---|---|---|---|---|---|---|
| RHS191 | uthSi17[myo-3p::MLS::GFP::unc-54 3'UTR, cb-unc-119(+)] I; unc-119(ed3) III; | myo-3p | pCFJ352 | atp-1 | eg6701 | Chromosome I |
| AGD2883 | unc-119(ed3) III; uthSi90[myo-3p::MLS::mRuby::unc-54 3'UTR cb-unc-119(+)] IV; | myo-3p | pCFJ356 | atp-1 | eg6703 | Chromosome IV |
| RHS193 | uthSi84[vha-6p::MLS::GFP(65C)::unc-54 3'UTR, cb-unc-119(+)] I; unc-119(ed3) III; | vha-6p | pCFJ352 | atp-1 | eg6701 | Chromosome I |
| RHS192 | uthSi83[col-19p:MLS::GFP(65C)::unc-54 3'UTR, cb-unc-119(+)] I; unc-119(ed3) III; | col-19p | pCFJ352 | atp-1 | eg6701 | Chromosome I |

### *C. elegans* microscopy

Imaging of mitochondrial morphology was performed using either a Leica Thunder microscope equipped with a 63x/1.4 Plan Apo-Chromat objective, standard GFP and DsRed filter, Leica DFC9000 GT camera, a Leica LED5 light source, and run on LAS X software, or Leica Stellaris confocal microscope equipped with a white light laser source and spectral filters, HyD detectors, 63x/1.4 Plan ApoChromat objective, and run on LAS X software. The WLL was set to 85.00% maximum power, using a 485-nm laser line at 3.00% intensity. The HyD S detector was configured to 490–590 nm with an analog gain of 25. Imaging was performed using unidirectional scanning over a 1,024 × 1,024 pixel area, corresponding to 82.01 × 82.01 µm, with 5 z-sections at a step size of 0.495 µm (z-dimension varied based on the sample size). Scanning parameters included a speed of 1,000 Hz, 2.25× zoom, line averaging of 2, and a pinhole size of 1 AU (95.5 µm). The Z-range was set from the first detectable GFP signal to its endpoint. Animals were placed in M9 solution directly on a glass slide, a cover slip is applied, and imaging is performed within 10 min of slide preparation (Kim et al, 2025). Quantification of mitochondrial morphology is performed using mitoMAPR (Zhang et al, 2019).

### *C. elegans* lifespan

All lifespan assays were performed on standard RNAi plates with HT115 bacteria at 20°C as previously described (Castro Torres et al, 2022). Animals were exposed to FUDR from the day 1 adult stage to eliminate progeny. Viability was scored every other day until all animals are scored or censored. Censorship is defined as animals that exhibit deaths unrelated to aging: vivipary (bagging), desiccation on the walls of the petri dish, intestinal leakage out of the vulva, etc. Survival curves were plotted, and log-rank statistical analyses were performed using Prism software. All statistical data for lifespans are available in Table S1.

### *C. elegans* Seahorse assay

A Seahorse assay was performed in day 1 adult animals synchronized using a standard bleaching protocol. Animals were collected off plates using M9, and bacteria were washed with M9 solution 3x using repeated centrifugation/aspiration. ~10–15 worms were pipetted into each well of a Seahorse XF96 cell culture microplate. Basal OCR was measured using a Fe96 sensor cartridge on Seahorse XFE96 Analyzer with 3-min mixing, 2-min wait, and 2-min measuring. Nonmitochondrial respiration rates were measured using 50 mM sodium azide as previously described (Haroon & Vermulst, 2019; Ng & Gruber, 2019). OCR was normalized for the number of worms.

### Statistical analyses

For all imaging experiments, quantification was performed using mitoMAPR and statistical analysis was performed using one-way ANOVA statistical testing. For lifespans, log-rank testing (Mantel–Cox) was performed. For Seahorse analysis, one-way ANOVA testing was used. All statistical tests were performed using Prism software. All experiments were performed across a minimum of three biological replicates.

# Supplementary Information

# Acknowledgements

J.K. is supported by the USC Provost Fellowship; M Vega is supported by 1R25AG076400; G Garcia is supported by T32AG052374 and R01AG079806-02S1 from the National Institute on Aging; and R Higuchi-Sanabria is supported by R01AG079806 from the National Institute on Aging and 2022-A-010-SUP from the Larry L. Hillblom Foundation. Some strains were provided by the CGC, which is funded by the NIH Office of Research Infrastructure Programs grant P40 OD010440. Some gene analyses were performed using WormBase, which is funded on a U41 grant HG002223.

## Author Contributions

J Kim: conceptualization, data curation, software, formal analysis, validation, investigation, visualization, and methodology.
N Dutta: conceptualization, data curation, software, formal analysis, validation, investigation, visualization, and methodology.
M Vega: data curation, formal analysis, and investigation.
A Bong: data curation, formal analysis, validation, and methodology.
M Averbukh: data curation, formal analysis, validation, visualization, and methodology.
R Aviles Barahona: data curation, formal analysis, validation, and methodology.
A Alcala: data curation, formal analysis, validation, and methodology.
JT Holmes: validation and visualization.
G Garcia: conceptualization, data curation, software, formal analysis, validation, investigation, visualization, and methodology.
R Higuchi-Sanabria: conceptualization, resources, data curation, formal analysis, validation, investigation, visualization, and methodology.

## Conflict of Interest Statement

The authors declare that they have no conflict of interest.

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
