## [Reviewer comments · Life Science Alliance]

Life Science Alliance

Cross comparison of strains used for mitochondrial imaging in *C. elegans* during aging.

Juri Kim, Naibedya Dutta, Matthew Vega, Andrew Bong, Maxim Averbukh, Rebecca Aviles Barahona, Athena Alcalá, Jacob Holmes, Gilberto Garcia, and Ryo Higuchi-Sanabria

DOI: <https://doi.org/10.26508/lsa.202403189>

Corresponding author(s): Ryo Higuchi-Sanabria, University of Southern California

Review Timeline:

Submission Date:	2024-12-23
Editorial Decision:	2025-01-22
Revision Received:	2025-04-04
Editorial Decision:	2025-05-06
Revision Received:	2025-05-22
Editorial Decision:	2025-05-23
Revision Received:	2025-05-23
Accepted:	2025-05-26

Scientific Editor: Tim Fessenden

Transaction Report:

January 22, 2025

Re: Life Science Alliance manuscript #LSA-2024-03189

Dr. Ryo Higuchi-Sanabria
University of Southern California
Leonard Davis School of Gerontology
3715 McClintock Ave GER312
Los Angeles, CA 90089

Dear Dr. Higuchi-Sanabria,

Thank you for submitting your manuscript entitled "Cross comparison of imaging strategies of mitochondria in *C. elegans* during aging." to Life Science Alliance. The manuscript was assessed by expert reviewers, whose comments are appended to this letter. We invite you to submit a revised manuscript addressing the Reviewer comments.

Thank you for this interesting contribution to Life Science Alliance. We are looking forward to receiving your revised manuscript.

Sincerely,

B. MANUSCRIPT ORGANIZATION AND FORMATTING:

Reviewer #1 (Comments to the Authors (Required)):

In the submitted work, "Cross comparison of imaging strategies of mitochondria in *C. elegans* during aging" by Juri Kim and colleagues, the authors use newly generated *C. elegans* strains, which contain stable single-copy transgenes expressing matrix localized green fluorescent protein (MLS::GFP), to evaluate changes in mitochondrial morphology as animals age. Mitochondria are known to fragment during aging and MLS-GFP, which is imported into the mitochondrial matrix, is a widely used marker for evaluating this phenotype (TOMM-20::mRFP is also noted as a common marker for mitochondria). The authors state that a potential disadvantage of using high-copy expression constructs (commonly used to express MLS::GFP and TOMM-20::mRFP) to evaluate mitochondrial aging is that importing so many proteins into the mitochondria may lead to mitochondrial stress. In the submitted work, the authors compare aging mitochondria morphology in their single-copy MLS::GFP constructs to that of multi-copy MLS::GFP and TOMM-20::mRFP and then use their single-copy strains to evaluate the effects of common experimental conditions on mitochondrial aging morphology. This work has the potential to highlight important factors that researchers should consider when evaluating mitochondrial morphology in aging animals. Unfortunately, the following oversights in preparing the manuscript have made it too difficult to evaluate this work in depth. Each of these problems should be addressed before resubmission.

- 1) The figures are not labeled. Though it appears they are ordered numerically, lacking the heading "Figure 1, Figure 2, etc...." makes it difficult to keep track of which figure is which while evaluating the work.
- 2) It appears that there was a mix-up in how the figures are referenced in the manuscript. In the body of the paper, where Figures 3 and 4 are referenced, it appears the authors intended to reference Figures 4 and 5 respectively. What appears to actually be Figure 3 (based on the order of the figures and figure legends) does not appear to be discussed anywhere in the paper.
- 3) Table S1 is referenced in the methods section. However, there does not appear to be a Table S1.
- 4) The authors note in the methods section that mitoMAPR was used to quantify mitochondria morphology. However, no quantification of mitochondria morphology is evident in the manuscript. This should be addressed since quantifying the phenotypes would help in evaluating the data. Furthermore, there is no indication of how many animals are represented by the images. This should also be addressed. Finally, the images should include scale bars.

Reviewer #2 (Comments to the Authors (Required)):

In this manuscript, Kim et al report three new transgenic *C. elegans* strains that can be used for the analysis of mitochondrial morphology. The three strains visualise the mitochondrial matrix in three different tissues, muscles, hypodermis and intestine. The strains contain a single copy of the respective transgene integrated at a defined genomic location. The authors analyse mitochondrial morphology in the three strains in wild-type and mitochondrial fission and fusion defective animals as well as during aging and on different food. In addition, they assess aging and basic physiological parameters such as movement. The strains complement recently published strains for the analysis of mitochondrial morphology in *C. elegans* by the Mair lab (Valera-Alberni et al, LSA, 2024). While I consider the strains useful for the *C. elegans* community, I have a few comments, which should be addressed.

1. The title of the manuscript is "Cross comparison of imaging strategies of mitochondria in *C. elegans* during aging" and in the abstract and summary, the authors state "To standardize imaging methods for mitochondrial morphology in *C. elegans*,..." and "The findings highlight technical considerations, imaging method standardization,...". Based on this, I expected the manuscript to be about imaging strategies and not about transgenic *C. elegans* strains. The title and these statements should therefore be changed throughout the manuscript in order to not mislead the reader.
2. The information provided about how the imaging was done is not sufficient. For example, what laser powers were used? Were standard laser power etc used for the different images provided in the figures?
In Materials and Methods, the authors state "Quantification of mitochondrial morphology is performed using mitoMAPR (Zhang et al, 2019)." However, at least as far as I can see, mitochondrial morphology was not quantified. Single images are shown for the different strains, conditions or ages but this data is qualitative. In addition, no statements are made about how many animals were analysed.

Finally, the authors state that "Animals were placed in M9 solution directly on a glass slide, a cover slip is applied, and imaging is performed within 10 minutes of slide preparation." Not to mount animals on agar pads but directly on glass slides is unusual and I am concerned about the animals' welfare. The fact that no methods were used to immobilize them indicates that they might have been dead at the time the imaging was done. This might explain some of the quite unusual morphologies observed (for example in the drp-1(RNAi) animals) and the differences in the controls for example the 'ev' controls in the three different strains in Fig. 1A vs Fig. 1B. This is a major concern. Can the authors clarify?

3. Related to the unusual structures that seem to be labelled in the MLS::GFP strains, what are the round structures labelled for example in Fig. 2C, vah-6 construct at day 13? Are those nuclei or vacuoles?

Reviewer #3 (Comments to the Authors (Required)):

Summary

The authors address the relevance of assessing mitochondrial morphology in *C. elegans* and describe several factors that are important for identifying issues for interfering with mitochondrial function using different methods. Using a set of fluorescent mitochondrial markers the authors convincingly show the benefits of single copy markers over the commonly used high copy markers. Their results contribute to a better use and understanding of mitochondrial research tools and their limitations in the important field of mitochondrial function.

The authors address important issues in *C. elegans* research like aging, use of different food sources, life and health span using a set of useful controls. Using these controls they are able to conclude the usefulness of food source OP1 when complemented with vitamin B12. They could also conclude that FUDR use in preventing offspring is only mildly interfering with physiology, allowing larger scale experiments than using the hand-picking method. The authors describe the limited physiological consequences of the techniques used, adding to the robustness of these type of experiments. At the same time they indicate the (mild) limitations of some of the techniques, like e.g. the possible risk of accumulation of red fluorescent probes.

We thank the editors and reviewers for an exceptionally timely and prompt review of the manuscript. We appreciate the positive comments about the work being a resource for the field and all the helpful comments to improve the manuscript. We believe that we have addressed all the reviewer concerns in this revised version of the manuscript.

Reviewer #1 (Comments to the Authors (Required)):

In the submitted work, "Cross comparison of imaging strategies of mitochondria in *C. elegans* during aging" by Juri Kim and colleagues, the authors use newly generated *C. elegans* strains, which contain stable single-copy transgenes expressing matrix localized green fluorescent protein (MLS::GFP), to evaluate changes in mitochondrial morphology as animals age. Mitochondria are known to fragment during aging and MLS-GFP, which is imported into the mitochondrial matrix, is a widely used marker for evaluating this phenotype (TOMM-20::mRFP is also noted as a common marker for mitochondria). The authors state that a potential disadvantage of using high-copy expression constructs (commonly used to express MLS::GFP and TOMM-20::mRFP) to evaluate mitochondrial aging is that importing so many proteins into the mitochondria may lead to mitochondrial stress. In the submitted work, the authors compare aging mitochondrial morphology in their single-copy MLS::GFP constructs to that of multi-copy MLS::GFP and TOMM-20::mRFP and then use their single-copy strains to evaluate the effects of common experimental conditions on mitochondrial aging morphology. This work has the potential to highlight important factors that researchers should consider when evaluating mitochondrial morphology in aging animals. Unfortunately, the following oversights in preparing the manuscript have made it too difficult to evaluate this work in depth. Each of these problems should be addressed before resubmission.

1) The figures are not labeled. Though it appears they are ordered numerically, lacking the heading "Figure 1, Figure 2, etc...." makes it difficult to keep track of which figure is which while evaluating the work.

We apologize for the confusion of this and have now embedded the figures into a compiled version of the manuscript. Note that we also have submitted unlabeled, high-resolution figure files and an editable word doc version of the manuscript without embedded figures as required by the journal alongside a non-compiled version of the manuscript, so take care to review the ones embedded into the manuscript for ease.

2) It appears that there was a mix-up in how the figures are referenced in the manuscript. In the body of the paper, where Figures 3 and 4 are referenced, it appears the authors intended to reference Figures 4 and 5 respectively. What appears to actually be Figure 3 (based on the order of the figures and figure legends) does not appear to be discussed anywhere in the paper.

3) Table S1 is referenced in the methods section. However, there does not appear to be a Table S1.

We apologize for this oversight and have now ensured all figure numbers are correct. We also apologize that Table S1 was missing and we have ensured that it is included in this revised version.

4) The authors note in the methods section that mitoMAPR was used to quantify mitochondria morphology. However, no quantification of mitochondria morphology is evident in the manuscript. This should be addressed since quantifying the phenotypes would help in evaluating the data. Furthermore, there is no indication of how many animals are represented by the images. This should also be addressed. Finally, the images should include scale bars. We have now included scale bars for all the figures. We have also included quantification using mitoMAPR for all of the figures with the exception of fusion and fission mutants, as previous

literature has shown that skeletonization of images, which is required for quantification via mitoMAPR (PMC6713509), results in artifacts when quantifying fusion and fission mutants (PMC10347554), which we also experienced and could not achieve robustly reproducible quantification of fusion fission mutants when using mitoMAPR.

Reviewer #2 (Comments to the Authors (Required)):

In this manuscript, Kim et al report three new transgenic *C. elegans* strains that can be used for the analysis of mitochondrial morphology. The three strains visualise the mitochondrial matrix in three different tissues, muscles, hypodermis and intestine. The strains contain a single copy of the respective transgene integrated at a defined genomic location. The authors analyse mitochondrial morphology in the three strains in wild-type and mitochondrial fission and fusion defective animals as well as during aging and on different food. In addition, they assess aging and basic physiological parameters such as movement. The strains complement recently published strains for the analysis of mitochondrial morphology in *C. elegans* by the Mair lab (Valera-Alberni et al, LSA, 2024). While I consider the strains useful for the *C. elegans* community, I have a few comments, which should be addressed.

1. The title of the manuscript is "Cross comparison of imaging strategies of mitochondria in *C. elegans* during aging" and in the abstract and summary, the authors state "To standardize imaging methods for mitochondrial morphology in *C. elegans*,..." and "The findings highlight technical considerations, imaging method standardization,....". Based on this, I expected the manuscript to be about imaging strategies and not about transgenic *C. elegans* strains. The title and these statements should therefore be changed throughout the manuscript in order to not mislead the reader.

We thank the reviewer for this comment. We have changed the title to "Cross comparison of strains used for mitochondrial imaging in *C. elegans* during aging" and have removed the sentence on technical considerations and imaging methods. These technical considerations and imaging methods are available elsewhere in a dedicated step by step protocol in our recently published JoVE (PMID: 39895615), and as the reviewer correctly pointed out, this study focused on specific strains and the scientific considerations behind using them. Thus, we have corrected the manuscript to highlight this focus.

2. The information provided about how the imaging was done is not sufficient. For example, what laser powers were used? Were standard laser power etc used for the different images provided in the figures?

This is a great question, and we have written into our methods what laser powers and other parameters were used in our study. However, it is also very important for researchers to optimize microscope settings for each experiment, as this is generally what is required to ensure robust data for each experiment – we have written this into the methods as well.

In Materials and Methods, the authors state "Quantification of mitochondrial morphology is performed using mitoMAPR (Zhang et al, 2019)." However, at least as far as I can see, mitochondrial morphology was not quantified. Single images are shown for the different strains, conditions or ages but this data is qualitative. In addition, no statements are made about how many animals were analysed.

We have included quantification using mitoMAPR for all of the figures with the exception of fusion and fission mutants, as previous literature has shown that skeletonization of images, which is required for quantification via mitoMAPR (PMC6713509), results in artifacts when quantifying fusion and fission mutants (PMC10347554), which we also experienced and could

not achieve robustly reproducible quantification of fusion fission mutants when using mitoMAPR. All sample sizes have been included in the figure legends.

Finally, the authors state that "Animals were placed in M9 solution directly on a glass slide, a cover slip is applied, and imaging is performed within 10 minutes of slide preparation." Not to mount animals on agar pads but directly on glass slides is unusual and I am concerned about the animals' welfare. The fact that no methods were used to immobilize them indicates that they might have been dead at the time the imaging was done. This might explain some of the quite unusual morphologies observed (for example in the *drp-1*(RNAi) animals) and the differences in the controls for example the 'ev' controls in the three different strains in Fig. 1A vs Fig. 1B. This is a major concern. Can the authors clarify?

This is a great point and we have now referenced that in our most recently published, dedicated step by step protocol in JoVE (PMID: 39895615), we have shown that worms placed on glass slides in M9 solution do not display aberrations in mitochondrial morphology until at least 20 minutes of being on the slide (figure below for convenience).

It should also be noted that we are not the only ones who have observed these strange, almost balled up looking mitochondria in *drp-1* loss of function. Previously published papers have also described this phenomenon, and call this a hyperfused structure (PMID: 29743663 and PMID: 29509812, figure copied below for convenience).

Abnormal ball-like mitochondrial structure of *drp-1* mutant animals (left, PMID: 29743663) and *drp-1* RNAi knockdown animals (PMID: 29509812).

However, to further add evidence that our methods are not flawed or inducing concerning artifacts for mitochondrial imaging, we have now performed similar timing experiments as our JoVE protocol where we compared worms directly on slides in M9 solution versus being on an agar pad. Of note, mitochondrial morphology is affected by the use of either sodium azide or tetramizole as shown above and in our JoVE paper, thus in this manuscript, we did not use any drugs or chemicals for any experiments and this made using agar pad particularly difficult since the method allows free movement of worms without sodium azide or tetramizole. Regardless, we still compared methods using agar pads to without and as our data very clearly show, there are no major differences in short-term imaging of worms on a glass slide compared to being on an agar pad, showing our methods are valid for short-term imaging (<15 min). Now this is added to the supplementary figure 1 and copied below for convenience.

3. Related to the unusual structures that seem to be labelled in the MLS::GFP strains, what are the round structures labelled for example in Fig. 2C, *vah-6* construct at day 13? Are those nuclei or vacuoles?

This is an astute observation by the reviewer. We can confirm these structures as an age-associated accumulation of autofluorescence that only exists in the intestinal MLS::GFP strain. We can confirm that this is autofluorescence as it is visible under the DAPI channel. Since this is a *vha-6p::MLS::GFP* transgenic animal, there should be no fluorescence in the DAPI channel except for autofluorescence. In this imaging, it is very clear that the circular structures are autofluorescence structures, which accumulate with age and many studies have shown there is an accumulation of autofluorescence in *C. elegans* with age (e.g., PMID: 27070172, PMID: 34099724, PMID: 20956318)

To further validate our claim, we have performed similar GFP-imaging in N2 wild-type worms lacking our transgene on day 5, 9, and 14, showing very similar ball-like structures in the GFP channel, indicating these are not mitochondria-related structures or artifacts of our mitochondrial strain.

Reviewer #3 (Comments to the Authors (Required)):

Summary

The authors address the relevance of assessing mitochondrial morphology in *C. elegans* and describe several factors that are important for identifying issues for interfering with mitochondrial function using different methods. Using a set of fluorescent mitochondrial markers the authors convincingly show the benefits of single copy markers over the commonly used high copy markers. Their results contribute to a better use and understanding of mitochondrial research tools and their limitations in the important field of mitochondrial function.

The authors address important issues in *C. elegans* research like aging, use of different food sources, life and health span using a set of useful controls. Using these controls they are able to conclude the usefulness of food source OP1 when complemented with vitamin B12. They could also conclude that FUDR use in preventing offspring is only mildly interfering with physiology, allowing larger scale experiments than using the hand-picking method. The authors describe the limited physiological consequences of the techniques used, adding to the robustness of these type of experiments. At the same time they indicate the (mild) limitations of some of the techniques, like e.g. the possible risk of accumulation of red fluorescent probes.

We thank the reviewers for the positive review of the manuscript and its utility.

May 6, 2025

Re: Life Science Alliance manuscript #LSA-2024-03189R

Dr. Ryo Higuchi-Sanabria
University of Southern California
Leonard Davis School of Gerontology
3715 McClintock Ave GER312
Los Angeles, CA 90089

Dear Dr. Higuchi-Sanabria,

Thank you for submitting your revised manuscript entitled "Cross comparison of strains used for mitochondrial imaging in *C. elegans* during aging." to Life Science Alliance. The manuscript has been seen by the original reviewers whose comments are appended below. As you will see there are no further requests from Reviewer 2. However, on assessing the revised manuscript for potential publication in Life Science Alliance, and in consultation with a member of our editorial board, we found that some important issues remain.

As requested by Reviewer 2, the revised work includes quantification of mitochondrial morphology by mitoMAPR. This provides important support for example images shown throughout the manuscript, and must be included as charts in main figures. We appreciate that this plugin cannot reliably quantify mitochondria with fission/fusion defects, however quantification must still be provided to support the example images shown. A simple quantification, in a manner of your choosing, must be included in the main figures to accompany these images.

Finally the clarification to Reviewer 2 in point 3 on autofluorescence is appreciated, however for the benefit of the reader these structures must be indicated in the figures where they appear with an explanation in the text and figure legend.

Our general policy is that papers are considered through only one revision cycle; however, given that the suggested changes are relatively minor, we are open to one additional short round of revision. Please note that a final decision will be made without additional reviewer input upon re-submission.

Please submit the final revision within one month, along with a letter that includes a point by point response to the remaining reviewer comments.

To upload the revised version of your manuscript, please log in to your account: <https://lsa.msubmit.net/cgi-bin/main.plex>
You will be guided to complete the submission of your revised manuscript and to fill in all necessary information.

B. MANUSCRIPT ORGANIZATION AND FORMATTING:

Sincerely,

Reviewer #2 (Comments to the Authors (Required)):

The authors have sufficiently addressed the concerns I had about the previous manuscript.

We thank the editors and reviewers for a very thorough and rapid evaluation of our revised manuscript. We appreciate the editors positive feedback and interest in the manuscript and understand the final points that were brought up. We have directly addressed both comments as requested by the editor and detailed below.

As requested by Reviewer 2, the revised work includes quantification of mitochondrial morphology by mitoMAPR. This provides important support for example images shown throughout the manuscript, and must be included as charts in main figures. We appreciate that this plugin cannot reliably quantify mitochondria with fission/fusion defects, however quantification must still be provided to support the example images shown. A simple quantification, in a manner of your choosing, must be included in the main figures to accompany these images.

We understand this important consideration and have now added thorough quantitative analyses in all the primary figures (with the exception of Figure 1 as Figure 2 provides thorough quantification of very similar conditions as Figure 1. Now each primary figure contains comprehensive quantification via mitoMAPR. We have also matched this quantification with a very comprehensive supplemental table 1 that includes all metrics of mitoMAPR beyond that is possible to put into a figure. Finally, we have expanded the discussion of this quantitative data, including some limitations when performing quantification of fusion/fission mutants, highlighting what can be concluded and what cannot due to potential artifacts when performing these quantitative measurements. We believe that this has now comprehensively addressed the editor concern, but also have made the manuscript stronger.

Finally the clarification to Reviewer 2 in point 3 on autofluorescence is appreciated, however for the benefit of the reader these structures must be indicated in the figures where they appear with an explanation in the text and figure legend.

This is a very important point, so we have put arrowheads into Fig. 3 when these structures are first discussed so that we can highlight what specific structures we are discussing. We also have a complete supplemental figure (Fig. S2) that highlights these structures and very clearly describe that these are autofluorescent structures seen in other studies as well, not restricted to our work or our reporters. We opted not to put arrowheads in every single figure that has intestinal mitochondrial images because this made the figures look overly complex and sloppy.

May 23, 2025

RE: Life Science Alliance Manuscript #LSA-2024-03189RR

Dr. Ryo Higuchi-Sanabria
University of Southern California
Leonard Davis School of Gerontology
3715 McClintock Ave GER312
Los Angeles, CA 90089

Dear Dr. Higuchi-Sanabria,

Thank you for submitting your revised manuscript entitled "Cross comparison of strains used for mitochondrial imaging in *C. elegans* during aging.". We appreciate the significant changes made to figures to include quantification of mitochondrial morphological features, which has substantially improved this work. We also note the inclusion of arrows in select images showing autofluorescence. However, the quantification of some features like length is self-evident, however it is not clear what "Mitochondrial Footprint" means which leaves the ratios shown in Figure 2 undefined. Please amend the results and figure captions to properly describe this metric. We would be happy to publish your paper in Life Science Alliance pending this change and final revisions necessary to meet our formatting guidelines.

- The abstract introduces the background that motivates this work but does not describe the main findings. Please rewrite the abstract to summarize the main findings with minimal background information.
- Please add the X and Bluesky handles of your host institute/organization as well as your own or/and one of the authors in our system.
- Please move your main, supplementary figure, and table legends to the main manuscript text after the references section.
- Please add an Author Contributions section to your main manuscript text.
- We encourage you to revise the figure legends for figure S5 such that the figure panels are introduced in alphabetical order.
- Please add callouts for Figures 3A-B; 4A-C; 5A-C; 6C, D; S4A-C and S5A-D to your main manuscript text.
- Please add scale bars to images in figures 1, S1, S2, S3 and S4.

A. FINAL FILES:

B. MANUSCRIPT ORGANIZATION AND FORMATTING:

Sincerely,

May 26, 2025

RE: Life Science Alliance Manuscript #LSA-2024-03189RRR

Dr. Ryo Higuchi-Sanabria
University of Southern California
Leonard Davis School of Gerontology
3715 McClintock Ave GER312
Los Angeles, CA 90089

Dear Dr. Higuchi-Sanabria,

Thank you for submitting your Resource entitled "Cross comparison of strains used for mitochondrial imaging in *C. elegans* during aging.". It is a pleasure to let you know that your manuscript is now accepted for publication in Life Science Alliance. Congratulations on this interesting work.

DISTRIBUTION OF MATERIALS:

Again, congratulations on a very nice paper. I hope you found the review process to be constructive and are pleased with how the manuscript was handled editorially. We look forward to future exciting submissions from your lab.

Sincerely,
